# Protocol for a systematic review on the methodological and reporting quality of prediction model studies using machine learning techniques

Constanza L Andaur Navarro [1,2] Johanna A A G Damen [1,2]
Toshihiko Takada [1], Steven W J Nijman [1], Paula Dhiman [3], Jie Ma [3],
Gary S Collins [3], Ram Bajpai [4], Richard D Riley,[4] Karel GM Moons,[1,2]
Lotty Hooft [1,2]

► Prepublication history and additional file for this paper are available online. To view these files, please visit the journal online (http://dx.doi.org/10.1136/bmjopen-2020-038832).

[1]Julius Center for Health Sciences and Primary Care, University Medical Center Utrecht, Utrecht University, Utrecht, The Netherlands
[2]Cochrane Netherlands, University Medical Center Utrecht, Utrecht University, Utrecht, The Netherlands
[3]Center for Statistics in Medicine, University of Oxford, Oxford, UK
[4]School of Primary, Community and Social Care, Keele University, Keele, UK

**Correspondence to**
Constanza L Andaur Navarro;
c.l.andaurnavarro@umcutrecht.nl

## ABSTRACT

**Introduction** Studies addressing the development and/or validation of diagnostic and prognostic prediction models are abundant in most clinical domains. Systematic reviews have shown that the methodological and reporting quality of prediction model studies is suboptimal. Due to the increasing availability of larger, routinely collected and complex medical data, and the rising application of Artificial Intelligence (AI) or machine learning (ML) techniques, the number of prediction model studies is expected to increase even further. Prediction models developed using AI or ML techniques are often labelled as a 'black box' and little is known about their methodological and reporting quality. Therefore, this comprehensive systematic review aims to evaluate the reporting quality, the methodological conduct, and the risk of bias of prediction model studies that applied ML techniques for model development and/or validation.

**Methods and analysis** A search will be performed in PubMed to identify studies developing and/or validating prediction models using any ML methodology and across all medical fields. Studies will be included if they were published between January 2018 and December 2019, predict patient-related outcomes, use any study design or data source, and available in English. Screening of search results and data extraction from included articles will be performed by two independent reviewers. The primary outcomes of this systematic review are: (1) the adherence of ML-based prediction model studies to the Transparent Reporting of a multivariable prediction model for Individual Prognosis Or Diagnosis (TRIPOD), and (2) the risk of bias in such studies as assessed using the Prediction model Risk Of Bias ASsessment Tool (PROBAST). A narrative synthesis will be conducted for all included studies. Findings will be stratified by study type, medical field and prevalent ML methods, and will inform necessary extensions or updates of TRIPOD and PROBAST to better address prediction model studies that used AI or ML techniques.

**Ethics and dissemination** Ethical approval is not required for this study because only available published data will be analysed. Findings will be disseminated through peer-reviewed publications and scientific conferences.

### Strengths and limitations of this study

► This protocol increases transparency to the methods and definitions used in our review and that are applied to develop prediction model studies using artificial intelligence or machine learning.
► The systematic review will provide an overview and critical appraisal of the methodological and reporting quality, and risk of bias of prediction model studies using machine learning.
► The findings of this review will provide the needed evidence for the development of tailored methodological and reporting guidelines for prediction model studies based on machine learning techniques.
► We will build a sensitivity search strategy by using terms related to machine learning techniques, as well as conventional prediction techniques.
► Language restriction to English might exclude additional studies published in other languages.

**Systematic review registration** PROSPERO, CRD42019161764.

## INTRODUCTION

Clinical prediction models aim to estimate the individualised probability that a particular outcome, for example, condition or disease, is present (diagnostic models) or whether a specific outcome will occur in the future (prognostic models).[1–4] Studies addressing the development, validation and updating of prediction models are abundant in most clinical domains. For example, in cardiovascular disease, more than 350 prediction models have been developed and only a few have been validated.[5] Moreover, systematic reviews have shown that, within different medical domains, the methodological and reporting quality of prediction model studies is suboptimal.[6–10] Due to the increasing availability

of larger, routinely collected and complex medical data, and the rising application of Artificial Intelligence (AI) or machine learning (ML) techniques for clinical prediction, the number of prediction model studies is expected to increase even further.

ML can be described as techniques that directly and automatically learn from data without being explicitly programmed for that task, and often without any prior assumption.[11–13] Thus, ML relies on patterns and inferences from the data itself. A perceived advantage of ML over conventional statistical techniques is its ability to analyse 'big', non-linear and high-dimensional data, and thus its ability to model complex associations and scenarios. Due to the novelty, diversity, flexibility and complexity of ML techniques, ML-based prediction model studies are often considered as uninterpretable for many users. Inadequate reporting of, for example, data sources, study design, modelling processes, number of predictors and other data assumptions, makes prediction models developed with ML techniques published in medical journals difficult to interpret and to be validated by other researchers, creating barriers to their use in daily clinical practice.

Complete reporting is essential to judge the validity of any prediction model as it facilitates: study replication, independent validation of the prediction model, risk of bias assessments, interpretation of the results, meta-analysis of prediction models, and the judgement of the value and applicability of such model in real clinical settings for individualised predictions.[14] While complete reporting reveals the strengths and limitations of a prediction model, it also enhances the use and implementation of prediction model in clinical practice. The 'Transparent Reporting of a multivariable prediction model for Individual Prognosis Or Diagnosis (TRIPOD)' statement has been available since 2015, providing a checklist of 22 items considered essential for informative reporting of diagnostic or prognostic prediction model studies.[15 16] Similarly, the Prediction model Risk Of Bias ASsessment Tool (PROBAST) was published in 2019 to guide the critical appraisal of prediction model studies.[17 18] PROBAST provides signalling questions to facilitate both the applicability and risk of bias assessment of prediction model studies across four domains: participants, predictors, outcome and analysis. This assessment can only be correctly implemented if prediction model studies are properly reported. Although TRIPOD and PROBAST both covered all types of prediction modelling studies, including those using ML techniques, their focus was on regression-based modelling. The challenges and necessity for reporting and quality assessment guidelines in the AI/ML field have been addressed by several authors and this has led to initiatives, such as Consolidated Standards of Reporting Trials-AI (for randomised controlled trials), and Standard Protocol Items: Recommendations for Interventional Trials-AI (for clinical trial protocols). Similarly, for prediction model studies using ML, TRIPOD-ML and PROBAST-ML have been announced.[19–21]

To improve the quality, transparency and usability of ML-based prediction models in medicine, it is important to explore the current use and reporting of ML techniques in prediction model studies, to evaluate the methodological conduct and risk of bias using PROBAST, and assess the adherence to TRIPOD by performing a comprehensive systematic review.[3 15–18 22]

### Study aim

The primary aim of this systematic review is to evaluate the reporting and the methodological conduct of studies reporting on prediction models developed with supervised ML techniques, across all medical fields. Specific objectives are to:

1. Evaluate the reporting quality of prediction models developed using ML techniques based on TRIPOD.
2. Assess the methodological quality and the risks of bias in prediction model development or validation studies using ML techniques based on PROBAST.
3. Identify key and emerging concepts for the development of tailored adaptations or extensions of both TRIPOD and PROBAST.

## METHODS AND ANALYSIS

Our systematic review protocol was registered with the International Prospective Register of Systematic Reviews (PROSPERO) on 19 December 2019 (CRD42019161764). This protocol was prepared using the Preferred Reporting Items for Systematic Reviews and Meta-Analysis Protocols (PRISMA-P) 2015 statement.[23]

### Eligibility criteria

Articles will be eligible for this review when describing primary studies on the development and/or validation of a multivariable diagnostic or prognostic prediction model with at least two predictors, using any supervised ML methodology within all medical fields, and published between January 2018 and December 2019. This last inclusion criterion is to obtain the most contemporary sample of articles that would reflect the current practices of applied methods in the ML prediction model field. We will include studies with any study design and data source, all patient-related health outcomes, all outcome formats and restricted to humans only. Further details about inclusion criteria are given in table 1.

Articles will be excluded from this review when reporting models that make predictions for enhancing the reading of images or signals (rather than for prediction of health outcomes in individuals), or use only genetic or molecular markers as candidate predictors. Furthermore, prognostic factor studies, secondary research, conference abstracts and studies for which no full text is available will also be excluded. The search will be restricted to articles available in English only. Further details about exclusion criteria are given in table 2.

**Table 1** Definition of inclusion criteria

| Inclusion criteria | Definition |
|---|---|
| Any study design | Articles that report the development and/or validation of a prediction model based on experimental studies or observational studies. This includes randomised controlled trials, prospective and retrospective cohort, case–control studies and case–cohort studies. |
| Using at least 2 predictors for risk prediction | Articles that report the development and/or validation of a prediction model using at least two predictors. Articles that use imaging or speech parameters as structured data plus other predictors such as clinical, demographics, histological and genetic risk scores features will be included. |
| Any data sources | Articles that report the development and/or validation of a prediction model using any structured data source, for example, electronic medical records, administrative claims data and individual patient data meta-analysis data. |
| Any supervised ML technique | Articles that report the use of any ML technique for development and/or validation of a prediction model. We will consider as a ML technique, a statistical technique based on advanced computational capacity and lower human intervention. More specifically, we will focus on supervised ML techniques. |
| Patient health-related outcomes | Articles that report the development and/or validation of a prediction model whose main outcome is on an individual patient level. We will include articles assessing diagnosis, prognosis and health services performance, such as length of stay or triage assessment. |
| All outcome measures format | Articles that report the development and/or validation of a prediction model whose main outcome has one of the following formats: continuous, binary, ordinal, multinomial and time-to-event. |

ML, machine learning.

## Information sources

A literature search will be systematically applied in one major public-available electronic medical literature databases from 01 January 2018 to 31 December 2019.

## Search strategy

The search strategy was built using keywords including ML-related terms (ie, 'supervised learning', 'support vector machine', 'neural network'), prediction-related terms[24] (ie, 'risk', 'prognosis') and several performance measures for prediction modelling (ie, 'AUC', 'O:E ratio'). For search refinement, we selected 30 articles aligned with our inclusion/exclusion criteria to create a 'golden bullet' set. This set was analysed using SWIFT-Reviewer to obtain the most frequent words in the included articles by topic modelling.[25] In MedlinerRanker, the analysis of the included and excluded golden bullets articles allowed us to obtain the most discriminative words to

be considered in the search strategy.[26] The final search strategy is presented in online supplemental file 1.

## Study records
### Data management

Study record information including title and abstract from the searched online database will be imported into EndNote Citation Manager and Rayyan systematic review software.[27] These platforms will track and back up all activities when authors conduct the literature review process. Once eligible studies are identified, full-text articles will be downloaded for full-text screening and data extraction. Data items (below) will be extracted from the final included studies for review using Research Data Capture (REDCap) software.[28]

**Table 2** Definition of exclusion criteria

| Exclusion criteria | Definition |
|---|---|
| Images or signal studies | Articles that report the development and/or validation of a prediction model for enhancing the reading of images, pathological samples or signals. The purpose of these articles is to improve the accuracy of an instrument rather than providing a clinical outcome. |
| Only genetic and/or molecular predictors | Articles that report the development and/or validation of a prediction model using only genetic and/or molecular candidate predictors. These articles are often based on high-dimensional data and unsupervised ML techniques. |
| Prognostic factors studies | Articles that report the identification of prognostic factors associated with a clinical outcome in an individual. |
| Secondary research | Articles that report narrative reviews, systematic reviews about prediction model studies in a specific medical field. Guidelines, expert's opinions and letters to the editor will also be excluded. |
| Conference abstract | Articles that report the development and/or validation of a prediction model presented in a conference. Such articles, by definition, do not report all the information required for assessment. |
| Full text not available | Articles that report the development and/or validation of a prediction model for which full text is not accessible online. |

ML, machine learning.

## Selection process

Two researchers, from a group of seven (CLAN, TT, SWJN, PD, JM, RB, JAAGD), will independently screen the titles and abstracts to identify eligible studies according to the eligibility criteria. Two independent researchers, from the combination of the previous seven reviewers, will review the full text for potentially eligible articles; one researcher (CLAN) will screen all articles and six researchers (TT, SWJN, PD, JM, RB, JAAGD) will collectively screen a portion of the same articles for agreement. Disagreements between reviewers will be solved by consensus or consultation with a third investigator, if necessary (JAAGD). The study flow will be presented in a PRISMA flowchart.[29]

## Data collection process

We will perform a double data extraction for all included articles. Two reviewers will independently extract data from each article using a standardised data extraction form. One researcher (CLAN) will extract data from all articles and six other researchers (TT, SWJN, PD, JM, RB, JAAGD) will collectively extract data from the same articles. The data extraction form will be piloted on five papers and amended, if necessary. Disagreements in data extraction will be discussed between the two reviewers, and adjudicated by a third reviewer (KGMM, GSC, RDR or LH), if necessary. The authors of the articles will be contacted for further information and clarification, if needed. Data and records will be maintained by the lead investigator (CLAN) and stored on a shared secure platform for access by all investigators (REDCap).

## Data items

Data to be extracted will be informed by TRIPOD using the TRIPOD adherence guidance, PROBAST and the CHecklist for critical Appraisal and data extraction for systematic Reviews of prediction Modelling Studies.[15–18 22 30] Additional items specifically relating to ML techniques for prediction model purposes, will also be extracted.

Extracted data will include study design for the development and validation of the model, outcomes to be predicted, setting, the intended use of the prediction model, study population, data source, patient characteristics, total study sample size, number of individuals with the outcome, number of predictors (candidate and final), internal validation type, predictive performance measures (discrimination and calibration), number of models developed and the details of the ML technique used to develop each model (eg, technique, preprocessing, data cleaning, optimisation algorithm, predictors selection, penalisation techniques, hyperparameters, code, data availability and so on). This form will contain instructions for the reviewers on how to assess the models presented in the articles. For example, the number of models developed will be based on how many ML techniques were used, including if several hyperparameters are tuned. We will set a limit to the number of models for data extraction to 10. The number of predictors will be counted based on what is reported in the article and/or supplemental file. If not stated, the number of predictors will be reported as unclear. The final data extraction form is presented in online supplemental file 2.

## Outcomes

The primary outcomes of this systematic review are the adherence to the TRIPOD reporting guideline and the risk of bias assessed using PROBAST.[17 18 22]

## Assessment of risk of bias

The risk of bias of individual studies is one of our outcomes of interest and will be assessed using PROBAST.[17 18]

## Data synthesis

We will conduct a narrative synthesis of the extracted data. Data will be summarised using descriptive statistics and visual plots. Numbers and percentages will be used to describe categorical data about the reporting, methodological conduct and risks of bias of the studies. The distribution of continuous data, such as sample size and the number of predictors, will be assessed and described using mean and SD for normally distributed data and using median and 25th and 75th percentiles for non-normally distributed data. The risk of bias assessment will be summarised and graphically presented for each PROBAST domain and as an overall risk of bias judgement. Results will be stratified by study type (development with internal validation and/or external validation), medical field and prevalent ML techniques.

## Meta-bias(es)

Meta-bias will not be investigated in this study.

## Confidence in cumulative evidence

The strength of the body of evidence will not be assessed in this study.

## Amendments

Protocol amendments will be listed and made available on the PROSPERO registration. Date, description and rationale will be given for each amendment.

## ETHICS AND DISSEMINATION

Ethical approval is not required for this study because only available published data will be analysed. The findings of this systematic review will be published in an open-access journal to ensure access for all stakeholders and disseminated in various scientific conferences.

## Patient and public involvement

Not applicable.

## Ethics approval and consent to participate

Not applicable.

## Consent for publication

Not applicable.

## DISCUSSION

The use of ML has been increasingly recognised as a powerful tool to improve healthcare by enabling related professionals to make decisions based on the increasingly available and diverse sources of (bio)medical data. Particularly, ML-based prediction algorithms that are considered the key to unlock the increasingly available data sources, are intended to better inform real-time clinical decisions, support early warning systems and provide superhuman imaging diagnostics.[31] However, published research about this topic rarely provides adequate information about the final predictive model, and its estimates and performance. Even more scarce is research where the prediction model is accessible for patients and healthcare professionals alike. Hence, ML-based prediction model studies are often seen as uninterpretable. This aspect of ML techniques is problematic especially in medical diagnosis and prognosis, hampering the judgement of quality, clinical acceptance and implementation.

At present, there is a limited number of systematic reviews regarding the reporting and methodological quality of ML-based prediction model studies and their risks of bias.[32–34] In this systematic review, we will review across all medical fields, the current use of ML techniques in prediction model development, validation and updating studies, the methodological conduct and risks of bias using PROBAST, and the adherence to the reporting guideline using TRIPOD. Particularly, we will assess the extent to which risks of bias and reporting of ML-based prediction model studies match the current recommendations from TRIPOD and PROBAST,[22] and the implications of these results to update or extend them to TRIPOD-ML and PROBAST-ML.

So far, our findings should be considered within limitations. ML is a recently developed concept and without a clear scope yet. Therefore, a sensitive search strategy is hard to build, which may result in a large number of abstracts to screen at initial stages. Additionally, we are only able to include articles in English, which will underrepresent research available in other languages.

**Acknowledgements** The authors would like to thank and acknowledge the support of Rene Spijker, information specialist.

**Contributors** The study concept and design were conceived by CLAN, JAAGD, KGMM, LH, PD, GSC and RDR. CLAN, JAAGD, TT, SWJN, PD, JM and RB will conduct article screening and data extraction. CLAN will perform data analysis. All authors drafted this manuscript, revised it for important content and have provided the final approval of this version. CLAN, the corresponding author, is the guarantor of the review.

**Funding** GSC is funded by the National Institute for Health Research (NIHR) Oxford Biomedical Research Centre (BRC) and by Cancer Research UK programme grant (C49297/A27294). PD is funded by the NIHR Oxford BRC.

**Disclaimer** The funder was not involved in the development of this protocol.

**Competing interests** None declared.

**Patient consent for publication** Not required.

**Provenance and peer review** Not commissioned; externally peer reviewed.

**ORCID iDs**
Constanza L Andaur Navarro http://orcid.org/0000-0002-7745-2887
Johanna A A G Damen http://orcid.org/0000-0001-7401-4593
Toshihiko Takada http://orcid.org/0000-0002-8032-6224
Steven W J Nijman http://orcid.org/0000-0001-6798-2078
Paula Dhiman http://orcid.org/0000-0002-0989-0623
Jie Ma http://orcid.org/0000-0002-3900-1903
Gary S Collins http://orcid.org/0000-0002-2772-2316
Ram Bajpai http://orcid.org/0000-0002-1227-2703
Lotty Hooft http://orcid.org/0000-0002-7950-2980

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
