## [Reviewer comments · BMJ Open]

ARTICLE DETAILS

TITLE (PROVISIONAL)	Protocol for a systematic review on the methodological and reporting quality of prediction model studies using machine learning techniques
AUTHORS	Andaur Navarro, Constanza L.; Damen, Johanna; Takada, Toshihiko; Nijman, Steven W.J.; Dhiman, Paula; Ma, Jie; Collins, Gary; Bajpai, Ram; Riley, Richard; Moons, Karel; Hooft, Lotty

VERSION 1 – REVIEW

REVIEWER	Ben Gibbison University of Bristol, UK. None declared - although I am an author of one of the manuscripts in the reference list, I would not consider myself an academic competitor of this group.
REVIEW RETURNED	19-Apr-2020

GENERAL COMMENTS	This is a worthwhile piece of work and the manuscript is clear and the outcomes are defined. There are some methodological details I would explained in the discussion a bit further: 1. Why only 2018 - 2019? Is this because of the number of records. Why not look at studies published from after publication of TRIPOD in 2015? 2. Why only one database (pubmed). Why not other medical databases and given that we know that most ML papers of routinely collected data are published in something like BioRxiv
--

REVIEWER	KARTHIKEYAN H CHTIST(Deemed to be University), banglore, Karnataka, India
REVIEW RETURNED	09-May-2020

GENERAL COMMENTS	1. Data collection process, can you specify the standardized data extraction form used in your research 2. Can you add more results to validate your work
--

REVIEWER	Hellen Santos Carlos Chagas Institute, Oswaldo Cruz Foundation, Brazil
REVIEW RETURNED	18-May-2020

GENERAL COMMENTS	The authors aim to review methodological aspects of diagnostic and prognostic predictive models based on machine learning approaches. This is a worthwhile problem. Nonetheless, the systematic review method should be improved and more detailed. At first, the research question the systematic review will address was not clear stated. Additionally, it is important to clarify the
---

	rationale for the review in the context of what is already known. The criteria for eligibility need to be described in more detail. For example, I did not understand why you will exclude models based only on genetic variants and why you established limits 2018/2019 to conduct the search. Finally, I think CHARMS checklist and QUIPS could be used as a method for assessing risk of bias and for quality review of original articles.
--	---

REVIEWER	Po-Hsuan Cameron Chen and Ronnachai Jaroensri Google Health, USA
REVIEW RETURNED	19-May-2020

GENERAL COMMENTS	Overall summary The paper is a protocol article for a planned study that systematically reviews the methodological and reporting quality of machine-learning (ML) based prediction model studies. Given the recent increases in ML-based clinical diagnostic models, it is good to evaluate how well the current studies adhere to the commonly used reporting guidelines. A study like this might raise awareness of the use of these guidelines and hopefully increase the rigor of future studies. Major comments  - The authors selected TRIPOD and PROBAST as the guidelines for evaluation. However, as the authors mentioned, there are also other suitable guidelines, such as STARD, CONSORT, SPIRIT. It's worth also including them or at least add a discussion on why excluding these guidelines. - The authors referred to the used ML method being like "black box". The notion of "black box" here implies that the results are not reproducible. However, this is very different from the notion of "black box" in the ML community. In the ML community, "black box" means the model makes predictions in ways we don't fully understand, but the results are usually reproducible. - The inclusion and exclusion criteria aren't clear enough. It is essential to describe the intended study type that the inclusion and exclusion criteria try to capture. It is unclear why we need to limit the number of predictors, and it is unclear how these are counted. For many diagnostic models, the model output is a diagnosis instead of a clinical outcome; therefore, requiring patient health-related outcomes as the inclusion criteria seems too stringent. - For Table 2, it would be great if the authors can elaborate on the exclusion of imaging/signal studies. If I understand correctly, the authors meant excluding studies along the line of superresolution, while studies on medical imaging based diagnostic tools are still within scope. Is this a correct understanding? - It's not clear why the authors exclude genetic and molecular predictors, and prognostic factor studies. There are substantial machine-learning-based studies like this that fit the article titled "prediction model studies using machine learning techniques." - The data items list is long (L196-201). It could be helpful to distill aspects of reproducibility that the authors are hoping to capture and design the data collection around those. - L186: It would be good to include the result of the pilot study here. Minor comments
--

	 - I'm not exactly sure the study can be done in a timely manner to inform necessary extensions or updates of TRIPOD and PROBAST. - L93: ML and statistics are closely related and often are the same thing with different names. - L163: The final search strategy should be summarized in the main article, not in the supplementary. - L173: It could be helpful to include researchers who have specialized expertise in ML/AI if there isn't one already. As studies in the medical domain and ML domain are often done slightly differently while using different terminologies. An ML expert could help bridge the gap in communication between the two domains. - L200: How do you count the number of predictors, number of models developed, etc.? These are often not reported, and there could be variability in how each research count.
--	--

VERSION 1 – AUTHOR RESPONSE

Reviewer #1:

1. *Why only 2018 - 2019? Is this because of the number of records. Why not look at studies published from after publication of TRIPOD in 2015?*

Author's response: Thank you for this comment. When we started this project and wrote this protocol, December 2019, we aimed to review a most contemporary sample of articles (at the moment of our start) that would reflect the current practices in the ML prediction model field. Considering that this is a highly rapidly emerging field, we explicitly chose to search the most recent literature. And considering that it is a review of currently used methodology in the field (rather than a review of, e.g., the effectiveness of a certain intervention that would require the most complete possible sample of articles), we took a sample of articles that would give a good overview of used methods: methods reviews use the concept of theoretical saturation rather than ensuring that no article is being missed. Our initial search was therefore over the entire year 2019 yielding approximately 13,000 articles (Jan-Dec 2019). Later, we decided to extend our search to the last two years (Jan 2018-Dec 2019) to extend our definition of latest, current literature, retrieving nearly 25,000 articles. Given the purpose of our review (i.e. methods overview in the most recent ML prediction modeling studies and conserving the concept of theoretical saturation) we consider the number of articles published during the last two years appropriate to perform a current state-of-the-art methods review.

We have added the following sentences to the methods and analysis section (L144):

"This last inclusion criterion is to obtain a most contemporary sample of articles that would reflect the current practices of applied methods in the ML prediction model field to date."

2. *Why only one database (pubmed). Why not other medical databases and given that we know that most ML papers of routinely collected data are published in something like BioRxiv.*

Author's response: Thanks for this comment. We acknowledge that it would have been interesting to use also BioRxiv or medRxiv as it would enable us to obtain a very recent sample of articles in the ML field. Nonetheless, BioRxiv/medRxiv is a preprint service rather than a journal. Thus, articles submitted to BioRxiv/medRxiv are not peer-reviewed, edited, or typeset before being posted online. As one of our objectives is to evaluate the reporting quality and risk of bias of the included articles, we chose to include only articles that have gone through peer-review by experts from their specific fields, as major improvements can be done after this process and better reporting can be obtained. Regarding other medical databases, PubMed has available articles from specialized journals related to ML (i.e. IEEE, journal of health informatics, journal of medical internet research) which in most of

the cases, are also indexed in other databases. We therefore think the PubMed sample is representative for articles available in other databases as well.

Reviewer #2:

1. *Data collection process, can you specify the standardized data extraction form used in your research.*

Author's response: Thanks for this comment. We have made available the data extraction form, as supplemental file 2.

2. *Can you add more results to validate your work*

Author's response: To validate our work, we designed our review to ensure that the entire screening and data extraction is done in duplicate. Additionally, to enhance the validation and transparency of our review, we added to the current version our search strategy and data extraction form as supplemental files.

Reviewer #3:

1. *The authors aim to review methodological aspects of diagnostic and prognostic predictive models based on machine learning approaches. This is a worthwhile problem. Nonetheless, the systematic review method should be improved and more detailed. At first, the research question the systematic review will address was not clear stated. Additionally, it is important to clarify the rationale for the review in the context of what is already known. The criteria for eligibility need to be described in more detail. For example, I did not understand why you will exclude models based only on genetic variants and why you established limits 2018/2019 to conduct the search. Finally, I think CHARMS checklist and QUIPS could be used as a method for assessing risk of bias and for quality review of original articles.*

Author's response: We also refer to our answer to the editor's comment and to reviewer 2 comment 2, above and the added supplements to the article. Our review aim is to evaluate the reporting quality and methodological conduct of studies reporting on diagnostic or prognostic prediction models developed with ML techniques. So far, we are not aware of other papers who have addressed this explicit aim. Our aim is to focus on prediction models not on prognostic factors. Accordingly, we will evaluate reporting quality by using the Transparent Reporting of a multivariable diagnostic and prognostic prediction models for Individual Prognosis Or Diagnosis (TRIPOD) guideline. We will assess the methodological quality and the risks of bias of included studies using indeed the CHARMS checklist and PROBAST tool (and not the QUIPS tool as this focuses on prognostic factor studies only), respectively. We will incorporate the TRIPOD adherence assessment instrument developed by Heus et al. (2018), items from the CHARMS checklist and the PROBAST tool into our data extraction form. We will also include additional ML-specific questions (i.e. hyperparameter tuning).

Several articles have been published about the reporting quality of prediction modeling studies in general, which we referenced in this protocol (Heus et al. 2018 doi:10.1186/s12916-018-1099-2; Collins et al. 2014 doi:10.1186/1471-2288-14-40; Bouwmeester et al. 2012 doi:10.1371/journal.pmed.1001221; Collins et al. 2011 doi:10.1186/1741-9-103). However, no systematic review has specifically assessed the risk of bias and reporting quality in ML-based diagnostic and prognostic prediction model studies.

Prompted however by this good comment and the related comments, we have updated Table 1 and 2 with clearer definitions of the inclusion and exclusion criteria, respectively. Our rationale for including only studies published in 2018 or 2019 is explained in our response to the editor and to reviewer 1.

Reviewer #4:

1. *The authors selected TRIPOD and PROBAST as the guidelines for evaluation. However, as the authors mentioned, there are also other suitable guidelines, such as STARD, CONSORT, SPIRIT. It's worth also including them or at least add a discussion on why excluding these guidelines.*

Author's response: We thank the reviewer for this comment. The context in which the STARD, CONSORT and SPIRIT guidelines are mentioned in the manuscript are the initiatives, such as CONSORT-AI and SPIRIT-AI (L120). Our intention here was to acknowledge the future development of reporting guidelines for AI/ML in different fields such as clinical trials. The objective of this systematic review is to critically appraise prediction modelling studies, rather than intervention or diagnostic accuracy studies, thus TRIPOD and PROBAST are the most applicable in this context. We have now made this more clear in the Introduction:

"The challenges and necessity for reporting and quality assessment guidelines in the AI/ML field has been addressed by several authors and this has led to initiatives such as, CONSORT-AI (for randomized controlled trials), and SPIRIT-AI (for clinical trials protocols),. Similarly, for prediction model studies using ML, TRIPOD-ML and PROBAST-ML have been announced."

- 2. The authors referred to the used ML method being like "black box". The notion of "black box" here implies that the results are not reproducible. However, this is very different from the notion of "black box" in the ML community. In the ML community, "black box" means the model makes predictions in ways we don't fully understand, but the results are usually reproducible.*

Author's response: We apologize for unnecessary confusion. We have stated in the manuscript that due to the novelty, diversity, flexibility, and complexity of ML techniques; ML based prediction model studies are often considered as a "black box" for many users (L98). As the reviewer mentions the concept of 'black box' refers to ML techniques being uninterpretable. We have improved our manuscript with the following sentence (L98):

"Due to the novelty, diversity, flexibility, and complexity of ML techniques, ML based prediction model studies are often considered as uninterpretable for many users."

- 3. The inclusion and exclusion criteria aren't clear enough. It is essential to describe the intended study type that the inclusion and exclusion criteria try to capture. It is unclear why we need to limit the number of predictors, and it is unclear how these are counted. For many diagnostic models, the model output is a diagnosis instead of a clinical outcome; therefore, requiring patient health-related outcomes as the inclusion criteria seems too stringent.*

Author's response: We thank the reviewer for this suggestion. First, we have updated Table 1 and 2 with clearer definitions of inclusion and exclusion criteria, respectively. To highlight the study type we try to capture, we have added the following statement (L142):

"Articles will be eligible for this review when describing primary studies on the development and/or validation of a multivariable diagnostic or prognostic prediction model with at least 2 predictors, using any supervised ML methodology within all medical fields, and published in Jan 2018- Dec 2019."

Second, we did not limit the maximum number of included predictors in a model but did stipulate that a model had to include "at least 2 predictors" which is the minimum requirement for developing a multivariable prediction model. The only limitation regarding predictors is studies developing a prediction model only with unstructured and high dimensional data (i.e. images and genetic variants, known as 'omics'). Predictors will be counted based on what is reported in the articles and/or supplemental files as "number of predictors; variables; covariates; factors; features". If this is not stated in the article or supplemental files, we will score the number of predictors as unclear. We have now included the following statement in the methods section of the manuscript (L186):

"This form will contain instructions for the reviewers on how to assess the models presented in the articles. For example, the number of models developed will be based on how many ML techniques were used, including if several hyperparameters are tuned. We will set a limit to the number of models for

data extraction to 10. The number of predictors will be counted based on what is reported in the article and/or supplemental file. If not stated, the number of predictors will be reported as unclear. The final data extraction form is presented in supplemental file 2.”

Finally, we consider a diagnosis to be clinical outcome, therefore diagnostic prediction modelling studies are to be included, as they constitute a patient health-related outcome. For further clarification, we have included the following statement in Table 1:

“We will include studies assessing diagnosis, prognosis, and health services performance, such as length of stay or triage assessment”

- 4. For Table 2, it would be great if the authors can elaborate on the exclusion of imaging/signal studies. If I understand correctly, the authors meant excluding studies along the line of superresolution, while studies on medical imaging based diagnostic tools are still within scope. Is this a correct understanding?*

Author’s response: Thanks for this comment and we apologize for the confusion. The reviewer is correct. Medical imaging based diagnostic tools usually include demographic and clinical variables. Therefore, it would be of interest for our review and we will include studies that develop prediction models based on medical imaging in combination with demographic and clinical variables. However, if a study aims to improve the accuracy of assessing a diagnostic image based only on superresolution feature analysis (i.e. no demographic or clinical variables are included in the model), it will be excluded. Nonetheless, we have updated Table 1 and 2 with clearer definitions of inclusion and exclusion criteria, respectively. Specifically, we have included the following statement in Table 1:

“Articles that report the development and/or validation of a prediction model for enhancing the reading of images, pathological samples, or signals. The purpose of these articles is to improve the accuracy of an instrument rather than providing a clinical outcome.”

- 5. It’s not clear why the authors exclude genetic and molecular predictors, and prognostic factor studies. There are substantial machine-learning-based studies like this that fit the article titled “prediction model studies using machine learning techniques.”*

Author’s response: Thanks for this comment. There are substantial ML- based studies in the field of genetic and molecular predictors, known as ‘omics’, and prognostic factor studies. The first studies usually performed unsupervised ML techniques in high-dimensional data. We intent to assess diagnostic or prognostic prediction modelling studies based on supervised ML techniques with structured data. Therefore, we have updated Table 1 with the following statement:

“Articles that report the use of any ML technique for development and/or validation of a prediction model. We will consider as a ML technique, a statistical technique based on advanced computational capacity and lower human intervention. More specifically, we will focus on supervised ML technique.”

Prognostic factor studies and prediction modelling studies both seek to understand and improve future outcomes in people with a specific disease or health condition. There are, however, also differences between the aims of both types of studies. Prognostic factor studies *aim to identify factors associated with subsequent clinical outcomes in people with a particular disease or health condition*. Prognostic factor research aims to discover and evaluate factors that might be useful as modifiable targets for interventions to improve outcomes, building blocks for prognostic models, or predictors of differential treatment response. (PROGRESS 2: *PLOS Med* 2013, doi: [10.1371/journal.pmed.1001380](https://doi.org/10.1371/journal.pmed.1001380)). On the other hand, prediction modelling studies use multiple prognostic factors in combination to predict the risk of future clinical outcomes in individual patients. A useful model provides accurate predictions (PROGRESS 3: *PLOS Med* 2013, doi: [10.1371/journal.pmed.1001381](https://doi.org/10.1371/journal.pmed.1001381)). We want to focus on ML-based prediction modelling studies, specifically. Therefore, ML-based prognostic factor studies would not fit our research aim.

6. *The data items list is long (L196-201). It could be helpful to distill aspects of reproducibility that the authors are hoping to capture and design the data collection around those.*

Author's response: We acknowledge the need for reproducibility. As mentioned in the manuscript, our data extraction form was built based on checklists and reporting guidelines like CHARMS and TRIPOD. The aim of these guidelines is to achieve proper reporting in order to make study results reproducible. Additionally, in order to improve reproducibility, we will screen and extract data in duplicate. We have made available the data extraction form as supplemental file 2.

7. *L186: It would be good to include the result of the pilot study here.*

Author's response: We understand this request. Nevertheless, our objective with the manuscript is to report the methodology of the systematic review. As mentioned in L186, we intent to pilot the data extraction form to obtain an optimal level of agreement between reviewers and it is intended to be done in 5 articles, as reported in the manuscript. This pilot will be done after we have selected eligible references, which is not the case yet.

8. *I'm not exactly sure the study can be done in a timely manner to inform necessary extensions or updates of TRIPOD and PROBAST.*

Author's response: We acknowledge the recommendation from the reviewer. Several of the authors listed on this review have vast experience in systematic reviews and we have planned this review based on their previous experiences. Also, we have selected many reviewers for this project to make sure we can finish everything in a timely manner.

9. *L93: ML and statistics are closely related and often are the same thing with different names.*

Author's response: We fully agree with the reviewer. ML and statistics are closely related and often used interchangeable. Usually ML is described as statistical techniques within the spectrum of advanced computational capacity and lower human intervention. Sometimes, "ML" is used to describe advanced statistics and "statistics", to conventional statistics, thus referring to their novelty. But it seems that the expected development of statistical techniques would be towards making use of advanced computational power. Therefore, to set a boundary between both would be illogical as they refer to the same methodology. Through our representative sample of included articles, we expect to be able to describe this spectrum of diverse ML techniques.

10. *L163: The final search strategy should be summarized in the main article, not in the supplementary.*

Author's response: Thanks for this comment. Indeed, search strategies are usually summarized in the main article in published protocols. However, we built a sensitive search strategy with several different terms, and it is not possible to report the full search strategy in the main article due to its extensiveness. Nonetheless, we have provided the complete search strategy, as supplemental file 1. Additionally, we have added more detail regarding the search strategy to the manuscript (L160):

"The search strategy was built using keywords including ML-related terms (i.e. 'supervised learning', 'support vector machine', 'neural network'), prediction-related terms²⁴ (i.e. 'risk', 'prognosis'), and several performance measures for prediction modelling (i.e. 'AUC', 'O:E ratio')."

11. *L173: It could be helpful to include researchers who have specialized expertise in ML/AI if there isn't one already. As studies in the medical domain and ML domain are often done slightly*

differently while using different terminologies. An ML expert could help bridge the gap in communication between the two domains.

Author's response: We fully agree with the reviewer. Specialized expertise is very helpful in any type of research, particularly in this topic where different terminology is used. Several of the authors listed in the manuscript have experience with Machine Learning which we consider necessary for this systematic review.

12. L200: How do you count the number of predictors, number of models developed, etc.? These are often not reported, and there could be variability in how each research count.

Author's response: Thanks for this comment. Predictors will be counted based on what is reported in the articles and/or supplemental files as "number of predictors; variables; covariates; factors; features". If this is not stated in the article, we will score the number of predictors as unclear. We have now added the information requested to the methods section with the following statement as an example (L210):

"For example, the number of models developed will be based on how many ML techniques were used, including if several hyperparameters are tuned. We will set a limit to the number of models for data extraction to 10. The number of predictors will be counted based on what is reported in the article and/or supplemental file. If not stated, the number of predictors will be reported as unclear. The final data extraction form is presented in supplemental file 2."

As the reviewer mentions and to avoid any type of discrepancies, we will make sure that all reviewers are aware of how to count certain items by piloting the data extraction form with each one of them. Additionally, we have added instructions to the data extraction form as reminder.

VERSION 2 – REVIEW

REVIEWER	Ben Gibbison University of Bristol UK
REVIEW RETURNED	30-Jul-2020
GENERAL COMMENTS	The authors have addressed my concerns - although I would like them to acknowledge why they didn't examine those in the pre-print literature - which is where the vast majority of the ML literature lies. Whilst their rebuttal is valid. They should allude to this in the manuscript - as many readers will wonder why they didn't search these servers.